# Resilience, Perceived Stress from Adapted Medical Education Related to Depression among Medical Students during the COVID-19 Pandemic

**DOI:** 10.3390/healthcare11020237

**Published:** 2023-01-12

**Authors:** Boonyarit Chakeeyanun, Nahathai Wongpakaran, Tinakon Wongpakaran, Awirut Oon-arom

**Affiliations:** Department of Psychiatry, Faculty of Medicine, Chiang Mai University, Chiang Mai 50200, Thailand

**Keywords:** depression, resilience, COVID-19, medical education

## Abstract

The COVID-19 pandemic disrupted several routine teaching methods in clinical settings which causing psychological distress among medical students. This study aims to explore the association of perceived stress related to the adapted medical education, resilience, and depression among medical students at a medical school in northern Thailand during the COVID-19 outbreak. A total of 437 medical students were recruited in January 2022. Data were obtained using self-administered online questionnaires, including demographic data, perceived challenges in learning online classes, the Resilience Inventory (RI-9), and the Patient Health Questionnaire 9 (PHQ-9). Binary logistic regression analysis was employed to explore factors associated with depression (PHQ-9 ≥ 9). A total of 27% of the participants were identified as having depression. Logistic regression revealed that the presence of previous psychiatric conditions (ß = 2.80, *p* < 0.05), stress from lack of in-person contact with peers (ß = 1.48, *p* < 0.05), stress from lack of in-person communication with teachers (ß = 1.49, *p* < 0.05), and level of resilience (ß = 0.88, *p* < 0.001) were independently associated factors for depression among medical students. Level of resilience was negatively correlated with depressive symptoms (r = −0.436, *p* < 0.001). The rate of depression among medical students was higher during the pandemic. Resilience was associated with depression. Strengthening resilience might have an important implication for depression among medical students.

## 1. Introduction

In March 2020, the World Health Organization (WHO) declared a global pandemic because of the emergent coronavirus-2019 (COVID-19), which was highly contagious and easily spread from human-to-human via direct contact and droplets [1]. The infection rates of COVID-19 abruptly increased in part because around 80% of infected patients showed no symptoms [2]. In Thailand, the spread of COVID-19 also led to lockdown measures and limited access to almost every hospitality industry, including traveling and transportation. European countries also used lockdown measures to prevent the spread of COVID-19. This changed the lives of more than 250 million people in Europe, especially students in all educational systems [3]. School lockdown measures were utilized in several countries and were one of the most common policies to prevent the spread of the disease. This policy led teachers to change several aspects of their teaching–learning methods to avoid close physical contact; however, this method was reported to cause more psychological distress. Studies among Chinese students demonstrated that both college and university students who experienced lockdown measures have significantly higher rates of depression compared to the students who were on-site, with a prevalence of 9% and 12.2%, respectively [4,5].

Since COVID-19 has emerged, adapted medical education during COVID-19 (AMEC) has been adopted. This includes online classes, restrictions on the number of attendants in clinical work, flipped classrooms, hybrid classrooms, and student-led education. In Thailand, all medical schools began online classes in July 2020 [6]. A study reported that medical students have more stress concerning online study than the spread of disease because they thought they had fewer opportunities to complete routine practices and procedures that were required in the medical curriculum [7]. The results also reported that medical students preferred to study in the classroom because of environmental factors, such as peer and patient contact. Furthermore, online classes were reported to increase stress and depression among medical students. During the outbreak, medical students working as healthcare providers were at greater risk of contracting COVID-19 because of their increased exposure. Moreover, final-year medical students’ induction into the healthcare workforce has been expedited to meet the surging demands, which has resulted in increased incidences of depression and anxiety [8]. The rate of depression was associated with the students’ ability to adapt to web-based classes, level of resilience, and coping styles. The rate of depression among Chinese medical students during COVID-19 increased to 9.6% while studying in online classes [9]. Therefore, the adaptation of teaching–learning methods as well as the internal factors of medical students, such as resilience and coping styles, were associated with psychological distress, especially depressive symptoms. This might have an eminent impact on health care and educational performance in this population.

During the COVID-19 pandemic, resilience is a common psychological variable that has been studied [10,11,12]. Resilience refers to the capability to adjust to adverse events and keep a balanced state of mind [13]. Lower resilience creates vulnerability to abnormal responses during specific situations while higher resilience is a protective factor against abnormal responses [14]. Having high resilience is very essential during the COVID-19 pandemic since it is one of the most important protective factors against psychological distress, such as depression. Previous studies found that resilience might help medical students adapt to uncertainty and maintain mental well-being while having online classes during the COVID-19 pandemic [9,15]. Furthermore, it was evident that resilience played a role in attenuating depression in medical and nursing students [16,17].

Although medical students are considered to be key persons in the healthcare system as they will become doctors or future specialists, they are considered a vulnerable population that is encountering concerns about heavy schedule patterns, the need to maintain excellent grades and performance, which is related to people who will receive their care, and preparation for the professional license examination during the COVID-19 pandemic [18,19]. Some of them also face high parental expectations or low family income. Students are therefore more prone to stress because of the burdens of studying, examinations, financial issues, and family expectations [8,19]. Enhancing medical students’ mental health was a priority for medical schools. Medical students used joint campaigns and strategies to relieve stress during the pandemic: video chat, social media applications, and mindfulness/meditation. Stress-relieving exercises and self-care activities like these should be promoted [20].

Therefore, this study aims to assess the relationship between perceived stress related to AMEC, resilience, and depressive symptoms and to identify associated factors of clinically significant depressive symptoms among medical students during the COVID-19 outbreak in Thailand. We hypothesized that having a high level of resilience should predict a low level of depressive symptoms.

## 2. Materials and Methods

### 2.1. Study Design and Participants

This cross-sectional study was conducted by recruiting medical students aged 18 and over who were studying medicine, excluding bioengineering and medical informatics, at Chiang Mai University in January 2022, during the ongoing pandemic. The participants who did not have active status as medical students were excluded since they might not be able to report their perceived stress related to AMEC.

We obtained data from all participants through online-distributed questionnaires and utilized convenience sampling. We distributed the invitation to the study and the questionnaire through online platforms, including a community webpage for medical students and a private text message application within each academic year of medical students. The questionnaire link was systemically limited to only medical students at Chiang Mai University. The inclusion criteria were screened, and information regarding informed consent and details about the study were included on the first page of the questionnaire, before participants answered questions, to ensure that participants met the study’s criteria, understood it, and were willing to participate in the study. The online questionnaires consisted of three major parts. The first part included demographic data characterizing the participants (i.e., sex, age, previous history of physical and psychiatric disorders, and learning outcomes). The second part identified perceived stress related to AMEC, i.e., lack of in-person peers or teachers and online examination. Resilience and depressive symptoms were assessed by using the resilience inventory (RI-9) and the Patient Health Questionnaires-9 (PHQ-9), respectively.

The ethical approval for this study was obtained from the independent research ethics committee of Chiang Mai University. Each participant was informed about research objectives, study design, and potential risks in the study. Informed consent was obtained before each participant started the survey. All information provided was kept confidential at Chiang Mai University, and the authors were authorized to use the data.

### 2.2. Measurements

The Resilience Inventory (RI-9) is a self-rating measurement consisting of nine items focused on individual recovery after encountering changing situation. Each question was scored on a 5-point Likert scale. The items have a response format with 5 options, where “1” means the statement “does not describe me at all” and “5” means “it describes me very well”. The minimum score is 9 and the maximum score is 45. Higher scores indicate greater levels of resilience. The psychometric properties of this measurement were tested in a sample of 140 university students and demonstrated good reliability and internal consistency [21]. The person reliability is 0.86 using Rasch analysis and the Cronbach’s alpha value is 0.90 in this study’s sample was 0.91.The Thai version of the Patient Health Questionnaire (PHQ-9) was used to measure depressive symptoms. The Thai version of PHQ-9 is a self-reported nine-question screening scale that uses a 4-point Likert scale to screen depressive symptoms. The 9 items describing the intensity of depressive symptoms was rated from 0 (“not at all”) to 3 (“nearly every day”). Scores on the Thai version of PHQ-9 are from 0 to 27, with scores ≥9 indicating significant depressive symptoms [22,23]. Higher scores represent greater depressive symptoms. The sensitivity at the cut-off value of 9 or greater was 0.84 and the specificity was 0.77. The internal consistency of the PHQ-9 in this study was acceptable (alpha coefficient = 0.79).Perceived stress related to AMEC was measured by a set of questions that asked about the extent of stress an individual perceived from lacking opportunities to complete their clinical practices in laboratories and their clinical works, in-person contact with peers, in-person communication with teachers, and completing examinations online. Participants responded to each question using a 4-Likert type of scale, ranging from 0 (no impact), 1 (mild impact), 2 (moderate impact), to 3 (severe impact) [7,8,9].

Please see the details of these measurements in the Appendix A.

### 2.3. Statistical Analysis

Descriptive statistics (e.g., percentage, mean, standard deviation) were used to demonstrate demographic data (e.g., age, gender, grades in medical school). The participants with and without depression were defined by a PHQ-9 score ≥9. The Chi-square test, independent *t*-test, and Mann–Whitney U test were utilized to compare differences between groups with and without depression depending on the types of collected data and their distribution. The correlations between Resilience Inventory and total score of PHQ-9 were also computed using Pearson’s correlation analysis.

To determine the associated factors of depression among medical students, the authors selected only statistically different factors between the depressed and non-depressed groups for regression analysis. We use the univariable binary logistic regression to define associated factors of depression, and only factors with *p ≤* 0.2 were considered as independent variables of multivariable logistic regression analyses. All possible associated factors of depression (PHQ-9 ≥ 9) were firstly included and then excluded one by one using the stepwise backward manipulation method. Pseudo R2 was used to evaluate the ratio of the sum of squares explained by the regression model.

## 3. Results

There were 437 participants enrolled in the study. The average age was 21.4 (±1.66) years. Of the total responses, 249 (57%) were from male medical students. Of all participants, 88% reported no previous history of psychiatric disorder. Most of the participants (80.8%; 353/437) reported their family monthly income as more than the mean national income according to the National Statistical Office Thailand in 2021 (395 USD or around 15,000 Thai Baht). Of all participants, 52.4% were in their preclinical years while the others were in their clinical years. The average PHQ-9 score for depressive symptoms of the medical students was 6.49 ± 3.4 (min: 0–max: 27). Of the 437 medical students evaluated in this study, 118 (27%) reported PHQ-9 scores of ≥9, which indicated clinically significant depressive symptoms in this study, and they were referred to as the depressed group. Sociodemographic data comparing the non-depressed and depressed group are provided in Table 1.

With regard to the perceived stress related to AMEC, medical students mostly reported that their stress had a “mild impact” on the loss of opportunities to practice in laboratories and clinical work (n = 233, 53.3%), lack of in-person contact with peers (n = 190, 43.5%), lack of in-person communication with teachers (n = 178, 40.7%), and online examinations (n = 188, 43%). When comparing the depressed and non-depressed groups, all aspects of perceived stress, including the loss of opportunities for medical practice and experiments, lack of in-person contact with peers, lack of in-person communication with teachers, and online examination, were significantly different. (*p* < 0.01). In terms of resilience, the average RI-9 score was 34.5 (±6.22). The resilience index was statistically different between the depressed and non-depressed group (*p* < 0.001). The comparative data between the depressed and non-depressed group in perceived stress of AMEC and resilience score are shown in Table 2.

To explore the associations of independent variables with depression, univariable logistic regression analysis was performed using the PHQ-9 score as the dependent variable. This score divided medical students into two groups, with scores ≥ 9 categorized as depression. Univariable logistic regression revealed that there were five factors associated with depression, including the presence of a previous history of psychiatric disorder (OR = 3.64; *p* < 0.001; 95%CI: 0.68,1.90), stress from loss of opportunities for practice and experiments (OR = 1.70; *p* < 0.001; 95%CI: 0.25, 0.81), lack of in-person contact with peers (OR = 1.807; *p* < 0.001; 95%CI:0.34, 0.84), lack of in-person communication with teachers (OR = 1.95; *p* < 0.001; 95%CI: 0.39, 0.94), online examinations (OR = 1.54; *p* < 0.001; 95%CI: 0.20 to 0.67), and resilience scores (OR = 0.87; *p* < 0.001; 95%CI: −0.18, −0.099).

Multivariable logistic regression analysis was performed to determine specific factors associated with depression. The analysis revealed that the presence of a previous history of psychiatric disorder, lack of in-person contact with peers, lack of in-person communication with teachers, and resilience index score were independently associated factors for depression among medical students (Table 2). For the correlation test, Spearman’s correlation demonstrated that the resilience index score was negatively correlated with depressive symptoms (r = −0.436, *p* < 0.001).

Figure 1 shows the scatter plots and regression between RI and PHQ-9 scores. The correlation coefficient was r = −0.436, *p* < 0.001.

## 4. Discussion

This study demonstrates the significant relationship between the perceived stress related to AMEC and depression. Consistent with other related studies, our findings have confirmed the existence of psychological distress among medical students in this period [24,25,26]. Many studies have proposed the hypothesis that stress and depression are related to the adapted methods of learning. Since we are moving into the post-COVID era, the trend of medical education for both undergraduate and post-graduate medical students must be adapted into blended and integrated learning that incorporates online and offline learning. Sharing online resources for medical education and developments on teaching skills in clinical assessment is necessary for medical educators to maintain teacher–student and doctor–patient interpersonal relationship in medical education [27,28]. The relationship of resilience and depression in medical students found in this study would be a protective factors for dealing with ongoing challenges in changing medical practice and education.

In line with the related studies, the disruption in teaching–learning methods and medical services in medical schools, such as online learning, virtual demonstration of cases, and medical procedures, was reported to cause stress and depression among medical students. One explanation for this situation could be that some medical students preferred traditional education methods and failed to adapt to online classes, making it difficult for them to cope with the changes. However, the results from this study should be carefully interpreted because there is higher level of mental health concerns and knowledge among medical students. This higher awareness and disease-specific understanding would probably lead to bias in answering self-reported questionnaires by both over- and underestimating psychological reports [29,30].

Sedentary behaviors that are caused by long periods of online learning have been shown to be related to stress and depression [31]. Less human-to-human contact is associated with depression. This suggests that the students may suffer from social isolation both inside and outside the class, which might be an important factors related to depressive symptoms. Likewise, difficulty in reaching out for help from teachers or friends during the lockdown increases the risk of developing depression, especially for those who are already lonely or have a type of personality disorder [32,33,34].

As expected, the prevalence rate of depression was higher during this pandemic than before using the same measurement. A previous study in northern Thailand showed a depression rate at 21.1% prior to the COVID-19 pandemic, which is slightly higher than the rate of depression in this study (27%) [35]. However, the depression rate in this study was relevant to the estimated overall prevalence of depression in medical students from 41 countries worldwide based on a systematic review, which demonstrated the depression rate to be 31% (23–40%). The prevalence of depression among medical students in Asia was also reported to be lower, especially in China, while the highest depression rate was found in the USA [26,36,37]. The different rates between Asia and other continents might be due to differences in individual factors, such as the average age of medical students as well as their roles as medical students in those countries. In terms of age, only 0.3% of the medical students in the United States were less than 24 years old while most of the medical students in Asia were aged between 18–24 years [24,38]. The lower age of medical students in Asia might be related to having more support from their families, while the medical students in Western countries who have a higher average age might experience lower social support and higher debt burden [39].

Our findings on the association between higher depressive symptoms and the presence of previous psychiatric disorders were similar to a previous study that suggested that the presence of a psychiatric disorder, or a chronic disease, higher perceived stress, and social media addiction were associated with an elevation in depression scores in medical students during the COVID-19 pandemic [25,40]. This finding suggested the need for preventive intervention for medical students who are at risk from these conditions.

In terms of psychological factors, resilience was widely found to be related to mental health outcomes in not only medical students but also medical staff [41] and would be one of the management strategies that could to prevent adverse mental health during the pandemic [12].

In addition, we found that resilience was negatively correlated to depression. This was similar to previous studies which found that higher states of resilience were predictive of lower rates of depression, and that higher resilience was also a protective factor for depression [9,42]. Our finding highlighted that resilience was a preventive factor for mental health. Resilience might play a role in social support, perceived stress, and sleep quality in medical students [43,44].

Even though there are many theories underpinning resilience, one common feature is the ability of an individual to cope with difficult situations, such as COVID-19. This ability enables the student to tackle stressful situations and live their lives despite mental challenges [45,46]. The mechanisms of resilience that helps them confront the changes include self-control, mindfulness, tolerance, wisdom (problem-solving), and perseverance [45,47]. For example, students with a high level of resilience may spend time with family or engage in hobbies during the COVID lockdown rather than meet physically with their friends [48]. Part of resilience is perseverance and self-control which prevents burnout for medical students or doctors when working in stressful environments [49]. However, action has yet to be taken in skill-based training for medical students.

### 4.1. Clinical Implications and Suggested Future Research

In terms of application in medical education, interventions that strengthened resilience for medical students during the COVID-19 pandemic and before other disruptions in teaching–learning methods might have important implications for depression in medical students. Several potential interventions and skills building for enhancing resilience exist among the adult population. These include meaning–making, mindfulness, and interpersonal mindfulness-based problem solving [50,51,52]. In addition, cognitive behavioral therapy (CBT)-based interventions, mindfulness-based interventions or mixed interventions, and those combining CBT and mindfulness training, can be applied to foster resilience skills. These can be conducted either in individual or in-group, face-to-face or online [53,54].

Digital mental health interventions might be one method to alleviate mental health in college students, such as internet-based cognitive–behavioral psychotherapy and human support in the form of coaching [55]. This work is one of the few studies to date that analyzes perceived stress from changed teaching–learning methods, resilience, and depression in medical students. Moreover, we explored a significant number of variables previously reported in related studies, with valid and reliable measurements and covering bio-psycho-social factors.

In terms of future research, investigation on the constructive model of the relationship between resilience and depression should be considered, including mediation or moderation effects of other psychological factors. Furthermore, an exploration on efficacy and acceptability in any implicated resilience training interventions by follow-up on depressive states in conditions of social isolation would enhance the options for managing depression among medical students.

### 4.2. Strengths and Limitations

To the best of our knowledge, this research is one of the early studies addressing the role of resilience on depressive symptoms among medical students during COVID-19 in Thailand.

Our study has some limitations. First, this study is cross-sectional, which limits its ability to set up causal associations between variables. Second, it was investigated only in one medical school. Therefore, it does not represent all medical faculties. In addition, further research is needed to consider data collection from other medical schools in the same model of this study so that the results may be generalized.

## 5. Conclusions

The rate of depression among medical students was found to be higher during the COVID-19 pandemic. Resilience was negatively associated with depression. Strengthening resilience in the current COVID-19 pandemic might have important implications for depression in medical students.

## Figures and Tables

**Figure 1 healthcare-11-00237-f001:**
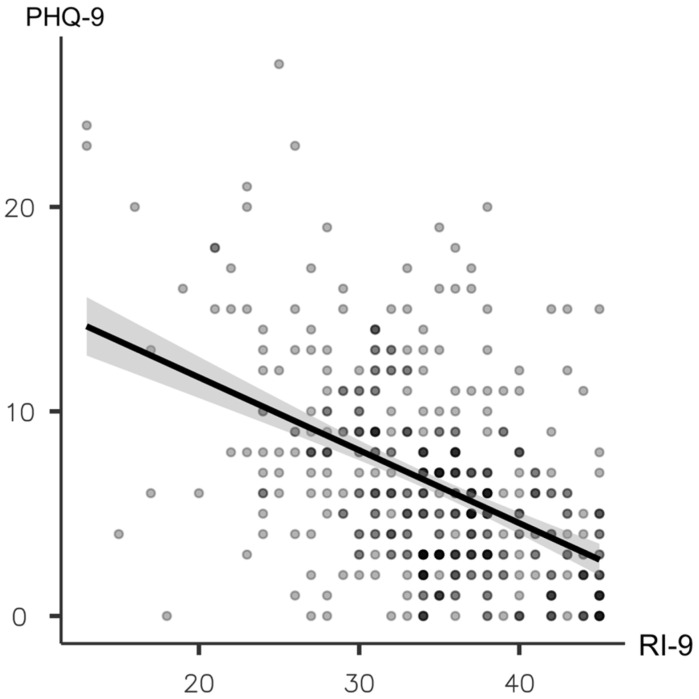
Pearson’s correlation test between Resilience Index (RI-9) and depressive symptoms (PHQ-9).

**Table 1 healthcare-11-00237-t001:** The comparative data between clinically significant depressed and non-depressed groups in sociodemographic data, perceived stress of AMEC, and resilience score.

Demographic Data	Depression (n = 118) N (%)	Non-Depression (n = 319) N (%)	*p* ^c^	*p* ^d^
Sex assigned at birth: Male	73 (61.86)	176 (55.17)	0.21 ^a^	
Presence of previous history of psychiatric disorder	92 (77.97)	296 (92.79)	<0.001 *^a^	<0.001 *
Income per month <395 USD ^e^	25 (21.19)	59 (18.50)	0.53 ^a^	
Age (year), Mean (SD)	21.3 (1.62)	21.4 (1.67)	0.77 ^b^	
Educational level			0.64 ^a^	
- Pre-clinical training	64 (54.24)	165 (51.72)		
- Clinical training	54 (45.76)	154 (48.28)		
- Living alone	79 (66.95)	184 (57.68)	0.079 ^a^	
Perceived stress of AMEC				
Loss of opportunities for clinical practice and experiments			<0.001 *	<0.001 *
Median (IQR)				
- No impact	16 (13.56)	74 (23.20)		
- Mild	57 (48.31)	176 (55.17)		
- Moderate	36 (30.51)	60 (18.81)		
- Severe	9 (7.63)	9 (2.82)		
Lack of in-person contact with peer. Median (IQR)			<0.001 *	<0.001 *
- No impact	21 (17.80)	85 (26.65)		
- Mild	36 (30.51)	154 (48.28)		
- Moderate	45 (38.14)	65 (20.38)		
- Severe	16 (13.56)	15 (4.70)		
Lack of face-to-face communication with teachers, Median (IQR)			<0.001 *	<0.001 *
- impact	32 (27.12)	160 (50.16)		
- Mild	56 (47.46)	122 (38.24)		
- Moderate	24 (20.34)	33 (10.34)		
- Severe	6 (5.08)	4 (1.25)		
Online examinations, Median (IQR)			<0.001 *	<0.001 *
- No impact	25 (21.19)	100 (31.35)		
- Mild	47 (39.83)	141 (44.20)		
- Moderate	28 (23.73)	62 (19.44)		
- Severe	18 (15.25)	16 (5.02)		
Resilience Inventory (RI-9), Mean (SD)	30.9 (6.18)	35.8 (5.68)	<0.001 *	<0.001*

^a^ Chi-squared tests, ^b^ Independent *t*-test, ^c^
*p*-value from difference test between participants with and without depression, ^d^
*p*-value from univariable logistic regression for depression, ^e^ mean income in Thailand 2021 from National Statistical Office Thailand, * statistically significant.

**Table 2 healthcare-11-00237-t002:** Multivariate logistic regression model for clinically significant depressive symptoms among medical students.

Variables	OR	*p* Value	95% CI
Previous history of psychiatric disorder	2.89	0.003	1.42 to 5.51
Lack of in-person contact with peers	1.47	0.014	1.08 to 2.02
Lack of in-person communication with teachers	1.49	0.021	1.06 to 2.10
Resilience Inventory (RI-9)	0.88	<0.001	0.84 to 0.92

## Data Availability

The datasets used and/or analyzed during the current study are available from the corresponding author upon reasonable request.

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
