# Peer review of "Resilience, Perceived Stress from Adapted Medical Education Related to Depression among Medical Students during the COVID-19 Pandemic"

_healthcare, 2023, doi:10.3390/healthcare11020237_

Round 1
Reviewer 1 Report
The work I have had the pleasure of reading concerns the relationship between depressive symptoms among Thai medical students and resilience skills.
The idea is good and original but some concepts should be explored:
Introduction:
1)If young medical students and young doctors were asked to change their work priorities and to be employed in roles other than their own specialization in the practices of contrasting the Covid-19 pandemic, this should be discussed in the introduction.
2)It should be investigated whether psychological help practices have been implemented by the hospital under investigation during the pandemic
Measurements:
1) line 116: are there any cut offs for scores? For example, a score of 20 is considered borderline, a score of 35 clinically significant, etc. ? If yes, please indicate
2)line 130: if possible add further details on the questionnaire such as considerations on the interpretation of the total score.
3)generally, details of the methods have to be implemented: description of the other screening questionnaires; method of administration; participant enrollment methods
Discussion:
1) The bias that doctors and medical students have in researching covid (due to their more in-depth knowledge of the diseases) should be mentioned in the discussion. This can impact psychological investigations involving medical students.
2)the literature on practices for implementing resilience should be cited in more detail. Future research could involve training on this construct followed by follow-up on depressive states in conditions of social isolation
3)in general, the concept of resilience and its relationship with the investigated constructs should be further explored
Author Response
Manuscript: Major Revision for manuscript entitled ‘RESILIENCE, PERCEIVED STRESS FROM ADAPTED MEDICAL EDUCATION RELATED TO DEPRESSION AMONG MEDICAL STUDENTS DURING THE COVID-19 PANDEMIC SITUATION’
Dear Editor,
We are appreciated that the editor provides us an opportunity to revise our manuscript, and that the reviewers provided us the useful comments. Please see below our point-by-point response to the reviewers’ comments.
Reviewer 1
The work I have had the pleasure of reading concerns the relationship between depressive symptoms among Thai medical students and resilience skills.
The idea is good and original but some concepts should be explored:
Response. Thank you.
Introduction:
- If young medical students and young doctors were asked to change their work priorities and to be employed in roles other than their own specialization in the practices of contrasting the Covid-19 pandemic, this should be discussed in the introduction.
Response. [line 56] During the outbreak, medical students working as healthcare providers are at greater risk of contracting COVID-19 due to increased exposure. Moreover, final-year medical students’ induction into the healthcare workforce has been expedited to meet surged demands resulting in increased incidences of depression and anxiety [8].
- It should be investigated whether psychological help practices have been implemented by the hospital under investigation during the pandemic
Response. [line 85] Enhancing medical students' mental health was a priority for medical schools. Medical students used joint campaigns and strategies to relieve stress during the pandemic: video chat, social media applications, and mindfulness/meditation. Several stress-relieving exercises and self-care activities like these need to be promoted [20].
Measurements:
- line 116: are there any cut offs for scores? For example, a score of 20 is considered borderline, a score of 35 clinically significant, etc. ? If yes, please indicate
- line 130: if possible add further details on the questionnaire such as considerations on the interpretation of the total score.
- Generally, details of the methods have to be implemented: description of the other screening questionnaires; method of administration; participant enrollment methods
We have described more details on methods as your suggestion in line 104:
Response. We distributed the invitation to the study and the questionnaire through online platforms, including a community webpage for medical students and a private text message application within each academic year of medical students. The questionnaire link was systemically limited to only medical students at Chiang Mai University. The inclusion criteria were screened, and information regarding informed consent and details about the study were included on the first page of the questionnaire before participants answered questions to ensure that participants met the study's criteria, understood, and were willing to participate in the study.
Answer for 1), and 2): Since there is no cutoff score for the Resilience Inventory (RI-9) and Perceived stress related to AMEC, we provide only the considerations on the interpretation of the total score as indicated in line 131:
For Resilience Inventory (RI-9): Higher scores indicate greater levels of resilience.
Discussion:
- The bias that doctors and medical students have in researching covid (due to their more in-depth knowledge of the diseases) should be mentioned in the discussion. This can impact psychological investigations involving medical students.
Response. [line 239] However, the results from this study should be carefully interpreted due to the higher level of mental health concerns and knowledge among medical students. This higher awareness and disease-specific understanding would probably lead to bias in answering self-reported questionnaires in both over- and underestimated psychological reports [29,30].
- The literature on practices for implementing resilience should be cited in more detail. Future research could involve training on this construct followed by follow-up on depressive states in conditions of social isolation
Response. [line 300] Several potential interventions and skills building for enhancing resilience among the adult population. These include meaning-making, mindfulness, and interpersonal mindfulness problem-solving [50-52]. In addition, cognitive behavioral therapy (CBT)-based interventions, (2) mindfulness-based interventions or mixed Interventions, those combining CBT and Mindfulness training, can be applied to foster resilience skills. These can be conducted either in individual or in group, face-to-face or online[53,54]
In terms of future research, the investigation on the constructive model of relationship between resilience and depression should be considered including mediation or moderation effects of other psychological factors should be considered. Furthermore, the exploration on efficacy and acceptability in any implicated resilience-training interventions by follow-up on depressive states in conditions of social isolation would enhance the options in management depression among medical students.
- In general, the concept of resilience and its relationship with the investigated constructs should be further explored
Response. [line 286] Even though there are many theories underpinning resilience, the common feature is an ability of an individual to cope with difficult situations, like during COVID-19. This ability enables the student to tackle stressful situations and live their lives generally despite mental challenges[45,46]. The mechanism of resilience that helps confront the changes includes self-control, mindfulness, tolerance, wisdom (problem-solving), and perseverance [45,47]. For example, students with a high level of resilience may spend time with family or engage in hobbies during the COVID lockdown rather than meet physically with their friends [48]. Part of resilience is perseverance and self-control that prevent medical students or doctors from burnout when working in stressful environments [49]. However, action has yet to be taken in skill-based training for medical students.

Reviewer 2 Report
A very interesting and important topic, and has a huge opportunity for future research.
Please describe the use of AMEC in the post-Covid period and the continued use of online learning today, and what are the advantages and disadvantages of this in the available literature.
Did the authors design the questions used to measure Perceived Stress in relation to AMEC?
If possible, please provide in supplement questions developed as well as the Resilience Inventory (RI-9) and Patient Health Questionnaires-9 (PHK-9). Update the table according to journal guidelines.
Please updated introduction and discussion.
Author Response
Manuscript: Major Revision for manuscript entitled ‘RESILIENCE, PERCEIVED STRESS FROM ADAPTED MEDICAL EDUCATION RELATED TO DEPRESSION AMONG MEDICAL STUDENTS DURING THE COVID-19 PANDEMIC SITUATION’
Dear Editor,
We are appreciated that the editor provides us an opportunity to revise our manuscript, and that the reviewers provided us the useful comments. Please see below our point-by-point response to the reviewers’ comments.
Reviewer 2
A very interesting and important topic and has a huge opportunity for future research.
Please describe the use of AMEC in the post-Covid period and the continued use of online learning today, and what are the advantages and disadvantages of this in the available literature.
We add this issue in discussion line 227:
Response. Since we are moving to post-COVID era, the trend of medical education for both undergraduate and post-graduate medical students must be adapted into blended and integrated learning from online and offline learning. Sharing online resource for medical education and development on teaching skill in clinical assessment is necessary for medical educator to keep teacher-student and doctor-patient interpersonal relationship in medical education. The relationship of resilience and depression in medical students found in this study would be one of the protective factors to deal with ongoing challenges in changing medical practice and education.
Did the authors design the questions used to measure Perceived Stress in relation to AMEC?
Response. No, we did not, but this questionnaire was developed by the authors from the literature we cited. Since these questions were developed in Thai, we also provide an unofficially English-translated version in supplementary section.
If possible, please provide in supplement questions developed as well as the Resilience Inventory (RI-9) and Patient Health Questionnaires-9 (PHQ-9). Update the table according to journal guidelines.
Response. The example and full version of the measurements (RI-9, PHQ-9, and the Perceived Stress in relation to AMEC) were provided in supplementary files.
Please updated introduction and discussion.
Response. We have updated introduction and discussion as suggested.
Thank you very much. We have addressed all the reviewers’ concern. We are looking forward to hearing your you soon.
Best regards,
AO and TW

Round 2
Reviewer 1 Report
The manuscript is now clearer and more coherent and it is possible to highlight its innovation